# The Effects of Artificial Intelligence Chatbots on Women’s Health: A Systematic Review and Meta-Analysis

**DOI:** 10.3390/healthcare12050534

**Published:** 2024-02-23

**Authors:** Hyun-Kyoung Kim

**Affiliations:** Department of Nursing, Kongju National University, 56 Gongjudaehak-ro, Gongju 32588, Republic of Korea; hkk@kongju.ac.kr

**Keywords:** anxiety, artificial intelligence, meta-analysis, systematic review, women’s health

## Abstract

Purpose: This systematic review and meta-analysis aimed to investigate the effects of artificial intelligence chatbot interventions on health outcomes in women. Methods: Ten relevant studies published between 2019 and 2023 were extracted from the PubMed, Cochrane Library, EMBASE, CINAHL, and RISS databases in accordance with the Preferred Reporting Items for Systematic Reviews and Meta-Analyses guidelines. This review focused on experimental studies concerning chatbot interventions in women’s health. The literature was assessed using the ROB 2 quality appraisal checklist, and the results were visualized with a risk-of-bias visualization program. Results: This review encompassed seven randomized controlled trials and three single-group experimental studies. Chatbots were effective in addressing anxiety, depression, distress, healthy relationships, cancer self-care behavior, preconception intentions, risk perception in eating disorders, and gender attitudes. Chatbot users experienced benefits in terms of internalization, acceptability, feasibility, and interaction. A meta-analysis of three studies revealed significant effects in reducing anxiety (I^2^ = 0%, Q = 8.10, *p* < 0.017), with an effect size of −0.30 (95% CI, −0.42 to −0.18). Conclusions: Artificial intelligence chatbot interventions had positive effects on physical, physiological, and cognitive health outcomes. Using chatbots may represent pivotal nursing interventions for female populations to improve health status and support women socially as a form of digital therapy.

## 1. Introduction

Artificial intelligence (AI) refers to the capability of a computer to exhibit human-like intelligence. An AI chatbot is a natural-language processing system that can independently engage in conversation with humans using artificial intelligence technology. Chatbots are software programs designed to interact in a manner similar to humans. They are distinguished by their use of machine learning and deep learning, which enable them to process data more efficiently and improve the quality of their responses through repeated learning [1]. Digital intervention platforms are taking various forms, such as mobile applications, telehealth, and web-based interventions. AI chatbots represent the latest innovation being employed in health interventions. These chatbots automatically generate knowledge from extensive language models and respond in a manner akin to humans, but with a greater volume and quality of information. This makes them an effective tool for delivering health education and interventions in the field of nursing [2].

Chatbots offer a range of advantages: they are engaging, entertaining, easy to use, informal, convenient, and interactive, without imposing an emotional burden on the user [3]. Additionally, they are cost-effective, foster a sense of social connection, provide aesthetic and hedonic pleasure, and possess a human-like tone, demeanor, and identity, all of which contribute to their potential to deliver digital therapy and mental health care [4]. Users tend to readily accept chatbots because they provide easy access to information, encourage ongoing use, are scalable, and have the capacity to store and retrieve knowledge [5]. However, there are potential drawbacks to using chatbots. These include the inability to verify the reliability of the information they provide, a human-like yet sometimes awkward interaction experience, varied outcomes based on the user’s skill in crafting prompts, and the risk of receiving simplistic, basic, or even misleading information. Moreover, chatbots lack an understanding of human complexity, which sets them apart from genuine human interactions [6]. An AI chatbot’s interaction with a user starts with an input, either text or spoken language, which is then analyzed using natural-language processing (NLP) to understand the intent and context. The chatbot determines the user’s intent through NLP analysis and generates a response either through rule-based methods for simple queries or machine learning for more complex interactions, improving over time with feedback [7]. It is anticipated that with NLP training, chatbots achieve capabilities that are nearly human-like or even surpass human quality. However, AI chatbots may lack the nuanced understanding and empathy of human healthcare providers. This limitation can be particularly significant in sensitive areas of women’s health, where emotional support and understanding are crucial [8]. They are expected to play a transformative role in the fields of health and education, serving as a new model for interaction between nurses and patients, as well as educators and learners [9].

The use of chatbots in health interventions has been explored in numerous recent studies. A thematic analysis of 37 papers focusing on chatbots in mental health revealed that these tools provide high-quality responses, characterized by their usability, responsiveness, understandability, acceptability, attractiveness, reliability, pleasantness, and content fidelity. Chatbots have demonstrated some effectiveness in interventions for conditions such as depression, anxiety, autism, addiction, post-traumatic stress disorder, schizophrenia, and stress [9]. Additionally, chatbots have been employed in interventions aimed at improving physical activity, diet, and weight loss. A systematic review of nine studies concluded that chatbots were effective in increasing physical activity, suggesting their potential as a physical health intervention [10]. Recent research has begun to explore chatbot interventions for specific populations, including pregnant women [4] and in the context of sex education [3]. The World Health Organization (WHO) identified cancer, infertility, reproductive health, violence, and mental health as crucial areas for improving women’s health globally. In 2023, the WHO reported that every two minutes, a woman still dies from complications related to pregnancy or childbirth [11]. AI chatbots can address specific challenges in women’s health by offering personalized health education, timely reminders for health-related tasks, and confidential consultations on sensitive issues [12]. However, there is a lack of systematic reviews focusing on chatbot interventions for women’s health. It is crucial to comprehensively investigate the impact of chatbot interventions on various aspects of reproductive health, such as pregnancy, childbirth, sexual health, female cancers, and women’s mental health. This would help determine the effectiveness of chatbots in these areas and inform future interventions. This paper provides the following major contributions:-It systematically reviews the impact of AI chatbot interventions on women’s health outcomes, providing a comprehensive analysis of current research.-It summarizes and synthesizes evidence on the effectiveness of chatbots in addressing key areas such as mental health, reproductive health, and chronic disease management among women.-It calls for further research into the development of culturally sensitive, user-friendly chatbot interventions to meet diverse health needs.

The aim of this study was to systematically review experimental studies analyzing the effects of AI chatbot interventions on women’s health. This review synthesized information on the topics addressed by chatbot interventions, the methods employed, and the outcomes of these interventions, presenting an integrated overview of the field. The systematic review synthesizes the literature on the impact of AI chatbots on women’s health, and the meta-analysis analyzes the effect size of the impact on women’s health outcomes.

## 2. Methods

### 2.1. Study Design

This study is a meta-analysis that systematically reviews experimental studies analyzing the impact of AI chatbot interventions on women’s health and analyzes the effect sizes of the interventions. The research question is “What is the impact of AI chatbot interventions on women’s health?” This study was conducted in accordance with the reporting guidelines for systematic reviews, as outlined in PRISMA [13] and the Cochrane Handbook for Systematic Reviews of Interventions version 6.3 [14].

### 2.2. Search Strategy

The researcher carried out a literature search from 30 September to 15 October 2023, using electronic information retrieval systems. Searches were conducted in the PubMed, Cochrane Library, EMBASE, CINAHL, and RISS databases, employing MeSH terms, Emtree terms, and natural language. The literature was identified based on inclusion criteria that followed the participant, intervention, comparison, outcome, setting, time-study design (PICOST-SD) framework [14]. The search strategy encompassed core databases and adhered to the guidelines of the Center for Occupational Safety and Health at the National Library of Medicine [15]. The inclusion criteria specified that studies must be (1) AI chatbot intervention studies applied to female users; (2) experimental studies designed as before-and-after studies, interrupted time series, non-randomized controlled trials, and randomized controlled trials; (3) studies that employed random, stratified, cluster, convenient, quota, or systematic sampling of participants; (4) studies that focused on women’s health, including cancer and reproductive, sexual, mental, and behavioral health; (5) studies published in either English or Korean; and (6) articles from peer-reviewed journals. The exclusion criteria ruled out (1) dissertations or conference presentations, (2) gray literature, (3) study protocols, (4) institutional reports or books, and (5) studies lacking statistical results. For the search, an advanced strategy was employed: ((“Text Messaging”[Mesh] OR “Instant Messaging”[Mesh]) OR (“Artificial Intelligence”[Mesh] OR “Chat*”)), ((“Text Messaging”[Mesh] OR “Instant Messaging”[Mesh]) OR (“Artificial Intelligence”[Mesh] OR “Chat*”)) AND “Nursing”, ‘chatbot women health’ OR ((‘chatbot’/exp OR chatbot) AND (‘women’/exp OR women) AND (‘health’/exp OR health)).

### 2.3. Literature Extraction

Literature extraction was performed using five databases: PubMed, Cochrane Library, EMBASE, CINAHL, and RISS. In total, 1592 articles were identified (899, 201, 427, 65, and 0, respectively) from 2011 to 2023. An additional 19 articles were sourced from reference lists and Google Scholar, bringing the total to 1611. After removing 257 duplicates, the researcher excluded 1314 articles based on their titles, resulting in 40 articles for further review. Upon reading the abstracts, the researcher selected three full-text articles for closer examination. Following a full-text review, the researcher excluded seven articles that did not focus on women’s health outcomes, five that were not experimental studies, eight that did not include female participants, and seven that did not involve chatbot interventions. Ultimately, 10 articles met the study criteria and were included in the review (Figure 1).

### 2.4. Quality Appraisal

The researcher assessed the quality of six out of the ten randomized controlled trials using the Cochrane Risk of Bias 2 (ROB-2) tool [16], which is designed to assess the quality of randomized controlled trials. This assessment was conducted independently by the researcher and a specialist in systematic reviews, with checks in place to ensure agreement. Inter-rater consistency was measured using the kappa coefficient; a coefficient of 0.80 or higher was considered acceptable, while a lower value prompted a consensus meeting for resolution [17]. The outcomes of the quality assessment were presented using traffic light charts and summary plots, generated by the Risk-of-bias VISualization (robvis) tool [18] (Figure 2).

### 2.5. Data Analysis

For a total of 10 papers, the researcher summarized (1) the topic of chatbot intervention, (2) the method of chatbot intervention, and (3) the effect of chatbot intervention, as detailed in a case report for each study tailored to the study’s objectives (Table A1). The case reports include the first author, publication year, country, chatbot title, intervention duration, setting, study design, participants, number of participants analyzed, and inclusion criteria (Table 1). Utilizing the study results, the researcher presents the primary and secondary outcomes, measurement scales, mean and standard deviation for both the experimental and control groups, effect size, and statistical significance (Table 2).

## 3. Results

### 3.1. Themes of Chatbot Interventions

The publication years of the articles were 2019 (A1, A2), 2020 (A3), 2021 (A4, A5), 2022 (A6), and 2023 (A7, A8, A9, A10). The countries of the articles were the United States (A2, A4, A7, A9), France (A1), Japan (A3), South Korea (A5), Zambia (A6), South Africa (A8), and Egypt (A10). The chatbot was given a proper name in nine articles (A1, A2, A4, A5, A6, A7, A8, A9, A10). The topics of the chatbots were as follows: breast cancer information (A1), psychological skills for young women (A2), preconception health (A3), eating disorders (A4), prenatal mental health (A5), HIV education and family planning (A6), genetic counseling for diagnosed breast cancer (A7), attitudes toward intimate partner violence (A8), prenatal health education for first-time mothers (A9), and self-care for chemotherapy side effects (A10) (Table 1).

### 3.2. Methods of Chatbot Interventions

The duration of the interventions varied, with the shortest being 20–30 min (A6) and the longest lasting up to 31 days (A8). Interventions took place in hospitals for six of the studies (A1, A2, A5, A7, A9, A10) and in community settings for the remaining four (A3, A4, A6, A8). Seven studies were randomized controlled trials (A1, A2, A3, A4, A7, A8, A10), and three had single-arm pre-post-test designs (A5, A6, A9). The participant demographics varied across studies: four targeted patients (A1, A2, A7, A10), three focused on young women (A3, A4, A7), two involved pregnant women (A5, A9), and one study included women of childbearing age (A6). The sample sizes also ranged widely, from as few as 15 participants (A5) to as many as 19,643 (A8) (Table 1).

### 3.3. Effects of Chatbot Interventions

The outcome variables were anxiety (A2, A3, A4), depression (A2, A4, A8), knowledge (A3, A6), satisfaction (A5, A7), answer or use rate (A1, A9), quality of information (A1), usage time (A2), emotion (A2), preconception intention (A3), eating disorder risk (A4), eating disorders (A4), internalization (A4), ease of learning (A5), acceptability (A6), feasibility (A6), interaction (A6), gender attitude (A8), intimate partner violence exposure (A8), unhealthy relationships (A8), physical effect (A10), psychological effect (A10), distress (A10), effectiveness of self-care behavior (A10), and usability (A10). The Patient Health Questionnaire was used to measure depression in two studies (A4, A8), and all other studies used their own instruments. 

The effects of the outcome variables showed that anxiety levels varied across studies (*p* = 0.09 and *p* < 0.001) (A2, A3, A4), and depression levels also varied (*p* = 0.07, < 0.001, and 0.01) (A2, A4, A8). Knowledge was found to be effective in both studies (*p* < 0.001) (A3,6), while satisfaction levels differed (*p* < 0.001 and 0.19) (A5, A7). The answer or use rate also differed, with values of 69% and 24.27%, respectively (A1, A9). Other effective outcome variables included the quality of information (*p* < 0.001) (A1), preconception intention (*p* < 0.001) (A3), risk of eating disorders (*p* < 0.001) (A4), eating disorders (*p* < 0.001) (A4), internalization (*p* = 0.001) (A4), ease of learning (*p* < 0.001) (A5), gender attitudes (*p* < 0.01) (A8), unhealthy relationships (*p* < 0.01) (A8), physical effects of unhealthy relationships (*p* < 0.001) (A10), psychological effects (*p* < 0.001) (A10), distress (*p* < 0.001) (A10), and the effectiveness of self-care behavior (*p* < 0.001) (A10). Outcome variables that were not found to be effective included ease of use (*p* = 0.24) (A5), positive and negative emotions (*p* = 0.97 and 0.82) (A2), and exposure to intimate partner violence (*p* > 0.05) (A8) (Table 2).

### 3.4. Results of Quality Appraisal

Seven randomized controlled trials (RCTs) (A1, A2, A3, A4, A7, A8, A10) were assessed for quality using the ROB 2 and robvis tools. In the first category, “bias arising from the randomization process”, bias was low except in two studies (A4, A7). In the second domain, “bias due to deviations from the intended interventions”, one study was rated as high (A1), while the others showed either no information or low bias. In the third domain, “bias due to missing outcome data”, all but three studies (A2, A3, A4) exhibited low bias. The fourth domain, “bias in measurement of the outcome”, presented high bias in three studies (A1, A3, A8). Similarly, the fifth domain, “bias due to the selection of reported outcomes”, indicated high bias in three studies (A1, A3, A8). In the overall evaluation, there was one study with a high risk of bias (A1), three with some concerns (A3, A4, A8), and three with a low risk of bias (A2, A7, A10) (Figure 2).

### 3.5. Effects of Chatbot Interventions on Anxiety

For the meta-analysis, the researcher utilized the Comprehensive Meta-Analysis V4 program to assess the effect size of three studies (A2, A3, A4) that examined the impact of chatbot intervention on anxiety. The researcher chose a fixed-effect model due to the limited number of studies, which precluded a reliable estimation of study variation [25]. The combined effect size was found to be −0.30, with a 95% confidence interval ranging from −0.42 to −0.18 (z = −4.91, *p* < 0.001) [26]. The heterogeneity analysis indicated no significant variation across the studies (I^2^ = 0%, Q = 8.10, *p* < 0.017) (Figure 3).

## 4. Discussion

This study conducted a systematic review of 10 articles examining the effects of AI chatbot interventions on women’s health and performed a meta-analysis on their impact on depression. The findings indicate that AI chatbot interventions are being utilized across a range of women’s health areas, although the number of such interventions remains limited and is still in the initial stages of development. The meta-analysis revealed a medium effect size for depression, suggesting that AI chatbot interventions can significantly influence women’s health. This study identified that AI chatbot interventions are being implemented in various domains of women’s health, including cancer care (A1, A7, A10), contraception (A3), violence (A8), sexually transmitted diseases (A6), dietary issues (A4), and pregnancy preparation (A5, A9). Overall, the results demonstrate that AI chatbot interventions are effective in enhancing both physical and mental health outcomes in the context of women’s health.

The studies reviewed were published from 2019 onward, indicating that the subject has seen active interest over the past five years, with the United States contributing four (A2, A4, A7, A9) out of the ten articles. Generative AI as we understand it today—particularly in the context of generating text, images, music, and other forms of media—began to gain prominence with the advent of more sophisticated machine learning models in the 2010s [27]. Breast cancer care emerged as the most frequently addressed topic in women’s health, represented in three out of the ten studies (A1, A7, A10). These studies explored the suitability of knowledge-generating chatbots for various functions: answering women’s questions about breast cancer (A1), providing genetic counseling for breast cancer (A7), and offering self-care strategies for managing chemotherapy side effects (A10). Chatbot interventions in cancer care offered the possibility for seamless integration into healthcare settings, complementing healthcare providers to improve patient care, streamline operations, and cut expenses [28].

The second most common intervention identified was prenatal education, covering topics such as mental health (A5) and general education (A9). The other five interventions each had distinct themes and targeted specific demographics: adolescents (A2, A4), women of childbearing age in need of contraceptive education (A3, A6), pregnant women (A5, A9), and women with partners (A8). The health conditions addressed included HIV (A6), breast cancer (A1, A7, A10), eating disorders (A4), and issues affecting women without other specific diseases. The findings suggest that chatbot interventions, which have traditionally focused on breast cancer and prenatal education, hold the potential to broaden their scope to encompass various aspects of women’s health throughout their life cycle. Privacy, security, and culturally sensitive approaches are essential for all women’s health topics. In the field of women’s health, the private nature of certain subjects, such as infertility, abstinence, and sexually transmitted diseases [24], may lead women to prefer the discretion of digital interventions over face-to-face interactions, which can be perceived as judgmental [3].

Physical health outcomes encompassed eating disorders (A4), intimate partner violence exposure (A8), and the effectiveness of self-care behavior (A10). Within these, interventions for eating disorders and breast cancer self-care proved to be effective. These interventions provided personalized education and improved access to real-time, high-quality information, showing usefulness and cost-effectiveness compared to traditional nurse-led education. This suggests that chatbots can play a significant role in physical self-care, particularly in managing chemotherapy-related side effects, by offering tailored support and information (A10). Another study presented the use of a teleassessment nursing chatbot application that enabled patients and families to assess general conditions, identify danger signs, and make informed decisions on utilizing health services during the COVID-19 pandemic [29]. Emotional health outcomes included anxiety (A2, A3, A4), depression (A2, A4, A8), satisfaction (A5, A7), emotion (A2), distress (A10), gender attitude (A8), and unhealthy relationships (A8). Of these, interventions addressing distress (A10), gender attitude (A8), and unhealthy relationships (A8) were found to be effective. Psychotherapy administered through a chatbot has been shown to significantly alleviate depressive symptoms, with a notable effect size (g = 0.54), indicating its effectiveness for adults dealing with depression or anxiety [30]. 

Cognitive health outcomes covered a range of areas such as knowledge (A3, A6), answer or use rate (A1, A9), quality of information (A1), usage time (A2), preconception intention (A3), ease of learning (A5), acceptability (A6), feasibility (A6), interaction (A6), internalization (A4), and usability (A10). Effective outcomes were observed in knowledge (A3, A6), quality of information (A1), preconception intention (A3), internalization (A4), ease of learning (A5), acceptability (A6), feasibility (A6), interaction (A6), and usability (A10). An AI chatbot was effective in encouraging healthy behaviors, with significant success in areas such as enhancing healthy living habits (40%), aiding in smoking cessation efforts (27%), supporting treatment and medication compliance (13%), and helping to decrease substance abuse (7%) [31].

The relatively small number of physical health outcome domains suggests that conversational AI chatbots may offer counseling comparable to that of health professionals, but may also provide inappropriate advice relative to human healthcare providers [19]. A significant limitation of AI, particularly when generated using large-scale language models, is the phenomenon of hallucination, which means it can complement but not surpass human interventions [19]. AI hallucinations could adversely affect decision making and pose ethical and legal concerns [32]. Chatbots currently lack the capability to fully grasp human complexity and do not match the level of human interaction. This was highlighted by a study showing that individuals still preferred to consult a doctor or nurse after interacting with a chatbot [6]. The proposed functional features of chatbots include functionality, efficiency, technical satisfaction, humanity, effectiveness, and ethics [7]. Consequently, while chatbots cannot substitute human involvement in physical health promotion, they can improve the efficiency of treatment when their use is validated by an expert.

Chatbot interventions have demonstrated effectiveness in improving cognitive outcomes in women’s health. The process of knowledge acquisition is intrinsically linked to individual perceptual experiences, highlighting the need for personalized educational strategies to enhance health outcomes for women. The results showed that chatbot technology has a medium-to-high impact on overall learning outcomes. In particular, chatbots significantly improved explicit reasoning, learning achievement, knowledge retention, and learning interest. However, some negative effects were observed in areas such as critical thinking, learning engagement, and motivation [33]. A review of 32 empirical studies on chatbot intervention revealed that chatbot technology had a medium-to-high effect on overall learning outcomes [33]. The cognitive benefits may stem from the fact that chatbot users experienced less cognitive load, were more productive, had lower frustration levels, and performed better overall, despite slightly longer task completion times [34]. Altogether, this meta-analysis determined that chatbot interventions had a moderate-sized effect on reducing anxiety in women’s health [17]. 

Another meta-analysis investigated the efficacy of digital interventions for all types of anxiety disorders. It reported a substantial pooled effect size (g = 0.80) in favor of digital interventions [35]. A chatbot designed to assess symptoms of depression and anxiety among university students was evaluated. The study revealed no significant differences in symptoms of depression or anxiety between the experimental group and the control group. However, there was a notable reduction in anxiety symptoms within the experimental group [36]. In another study, a psychoeducational chatbot aimed at helping university students manage stress and anxiety was interacted with an average of 78 times over the course of the study. This interaction led to a significant reduction in symptoms of anxiety and stress [37]. Chatbots are beneficial in mental health interventions for several reasons. One is that social support provided through conversation elicits positive emotions [20]. Another is that chatbots disseminate knowledge, thereby increasing the information available to users, enhancing their understanding, and reducing health-related anxiety. Additionally, the interactive feedback from chatbots can reduce stress because it is accessible, convenient, informal, and free from embarrassment. Chatbots engage in a human-like manner, offering empathy without human judgment and creating a sense of safety [3].

No chatbot interventions were found that studied the long-term effects of women’s health, but mental health, HIV, and cancer management are issues that warrant long-term interventions. The acceptability of an autonomous virtual agent designed to support self-management for patients with chronic diseases, which are long-term conditions that often co-occur with mental health problems such as anxiety and depression, remains to be determined. AI chatbots overcome the physical limitations of human therapists, such as the ability to operate without fatigue and to be available almost anywhere and anytime [38]. Schmidhuber et al.’s suggested improvement to reduce physical workload and cognitive demand was the implementation of text prediction features, which could help lower the effort required for typing [34]. The integration of AI-powered chatbots into the healthcare system promises to improve patient engagement, streamline administrative processes, and increase access to care, particularly in mental health support. However, realizing their full potential requires addressing challenges related to data privacy, accuracy, and the ethical implications of AI in healthcare [39]. Further research is needed to directly compare the effectiveness of AI chatbots with traditional healthcare interventions like face-to-face counseling, telehealth, and mobile applications, to better understand their potential and limitations in enhancing healthcare delivery.

AI chatbots are at the forefront of technology in behavior change interventions [34]. Understanding how human intervention can effectively induce behavior changes is crucial, especially in reducing anxiety. There is a need to explore the relationship between chatbot intervention strategies and their impact on women’s perceived barriers and social support for health outcomes [40]. Despite the rapid user growth of chatbots and their utility for health interventions, concerns have been raised about their decision-making accuracy, precision, and the ethical and legal implications [32]. AI and robotics are starting to be deployed in women’s health globally, emphasizing the disparity between developed and underdeveloped regions. This study advocates for an inclusive approach in technology design and implementation to ensure equitable healthcare access, highlighting the importance of multi-sectoral collaborations to foster innovation while mitigating risks [41]. Not all women may have equal access to the technology needed to interact with AI chatbots, especially in low-resource settings or among older populations. This digital divide can exacerbate existing health disparities. Outcome variables that were not found to be effective included ease of use (A5). AI chatbots developed without considering diverse cultural and linguistic contexts may not be effective for all women. Tailoring interventions to fit different cultural backgrounds and languages is essential but challenging [42]. These interventions must be applied to various women’s health issues through more interdisciplinary work to circumvent ethical problems [43].

### Limitations

This study has a few limitations. First, articles published after October 2023 were not included as they fell outside the search period. Despite the recent surge in chatbot interventions, this may have resulted in the exclusion of a significant body of research. Second, limiting the article search to English and Korean publications may have resulted in a geographical and cultural bias, excluding potentially valuable studies conducted in other languages. Third, the wide variety of outcome variables presented challenges for meta-analysis, which led to the decision to only conduct a meta-analysis of data related to anxiety. Fourth, the rapid development of AI technologies means that significant contributions to the literature could have been made just after the cutoff, potentially leaving out innovative approaches or critical evaluations of chatbot interventions. Fifth, the small sample sizes of the included studies may not provide sufficient power to detect significant differences or to ensure the representativeness of the results. 

## 5. Conclusions

Thus systematic review and meta-analysis of AI chatbot interventions in women’s health has illuminated the significant potential of these technologies to enhance healthcare outcomes. Chatbots have shown considerable promise in delivering health education, supporting mental health, managing chronic diseases, and providing targeted interventions across a spectrum of women’s health issues, including reproductive health, cancer care, and prenatal education. Notably, these interventions have been effective in improving both physical and mental health outcomes, with a particular emphasis on reducing anxiety, underscoring the value of integrating AI chatbots into healthcare strategies. Consequently, the use of chatbot interventions is recommended in mental health programs due to their proven effectiveness in alleviating anxiety. Given that chatbot interventions have demonstrated high feasibility, usability, and acceptability, they hold significant potential to become scalable interventions.

## Figures and Tables

**Figure 1 healthcare-12-00534-f001:**
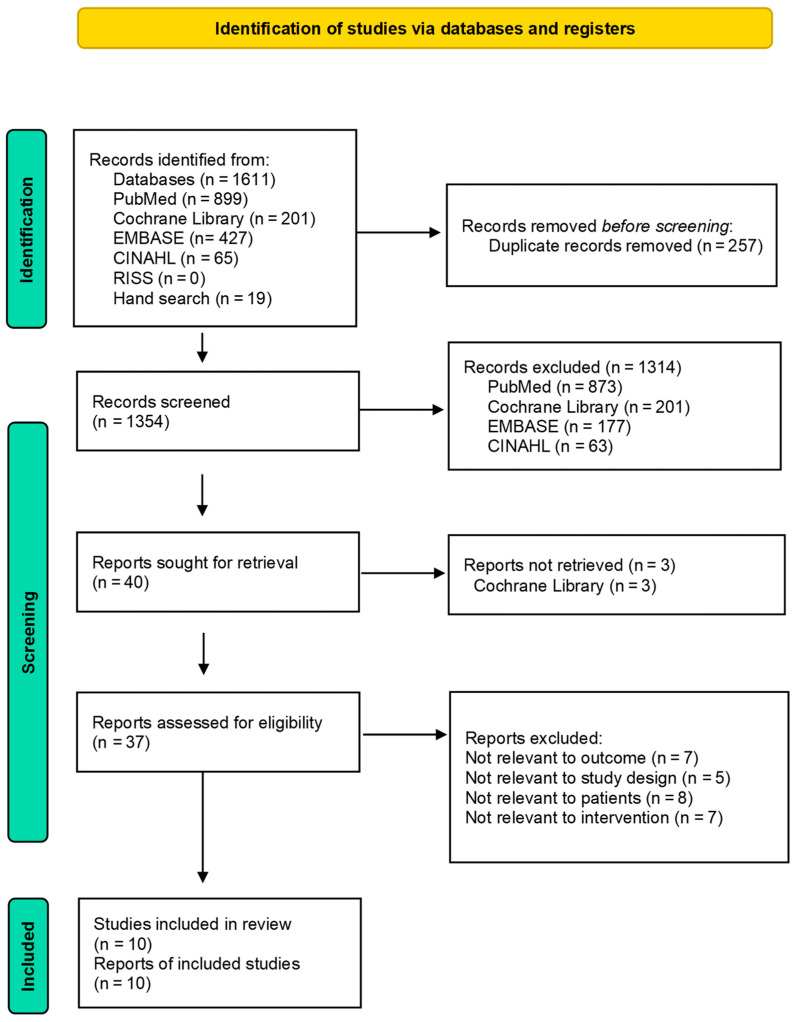
PRISMA flow diagram.

**Figure 2 healthcare-12-00534-f002:**
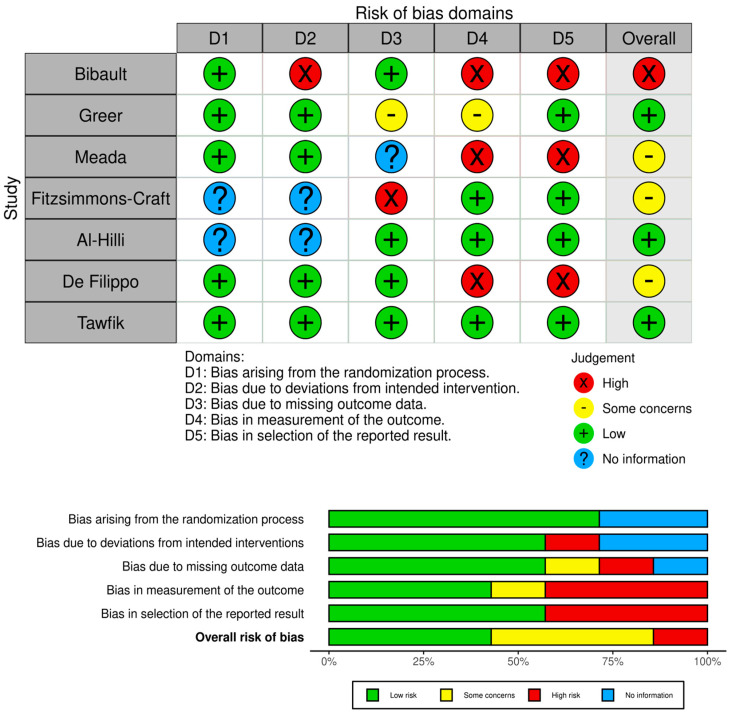
Risk of bias for the randomized controlled trials.

**Figure 3 healthcare-12-00534-f003:**
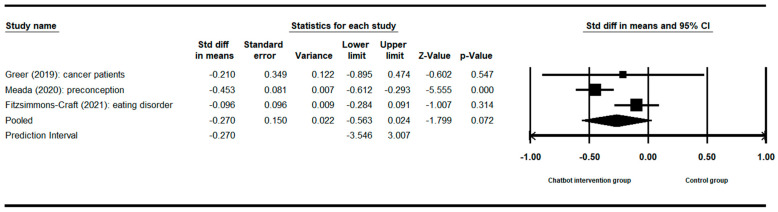
Forest plot of anxiety as an outcome [3,20,21].

**Table 1 healthcare-12-00534-t001:** Characteristics of the selected studies.

No	First Author	Publication Year	Country	Name of Chatbot	Theme	Intervention Time	Setting	Study Design	Participants	Number of Participants (Exp.–Cont.)	Inclusion Criteria
A1	Bibault [19]	2019	France	Vik	Breast cancer information	Four weeks	Hospital	Randomized controlled trial	Patients	142 (71–71)	Breast cancer patients
A2	Greer [20]	2019	USA	Vivibot	Psychological skill	Four weeks	Hospital	Randomized controlled trial	Patients	45 (25–20)	Young women aged 18–29 after cancer
A3	Maeda [3]	2020	Japan	Education Chatbot	Preconception health	Ten days, 574 sessions	Community	Three-armed randomized controlled trial	Young women	927 (309–309–309)	Young women aged 20–34
A4	Fitzsimmons-Craft [21]	2021	Zambia	Student Bodies	Eating disorder prevention	One month	Community	Randomized controlled trial	Young women	439 (207–232)	Young women aged 18–30
A5	Chung [4]	2021	South Korea	Dr. Joy	Prenatal mental health	13 days	Hospital	Single-group pre-post-test design	Pregnant women	15	Pregnant women with spouses
A6	Yam [5]	2022	USA	HIV Chatbot	HIV education and family planning	20–30 min	Community	Single-group pre-post-test design	Reproductive-age women	30	Women aged 15–49 in Zambia
A7	Al-Hilli [22]	2023	USA	Gia	Cancer genetic counseling	No information	Hospital	Randomized controlled trial	Patients	37 (19–18)	Breast cancer patients who received genetic counseling
A8	De Filippo [23]	2023	South Africa	Chatty Cuz	Intimate partner violence attitude	31 days	Community	Four-armed randomized controlled trial	Young women	19,643 (5891–5893–3930–3929)	Women aged 18–24 suffering intimate partner violence in South Africa
A9	Mane [24]	2023	USA	Rosie	Prenatal health education	No information	Hospital	Single-group pre-post-test design	Pregnant women	109	Primi-pregnant women aged over 14
A10	Tawfik [6]	2023	Egypt	Chemofree Bot	Chemotherapy self-care	45 min, 7–10 sessions	Hospital	Three-armed randomized controlled trial	Patients	150 (50–50–50)	Breast cancer patients undergoing chemotherapy

Cont = Control group; Exp = Experimental group; HIV = Human immunodeficiency virus; USA = United States of America.

**Table 2 healthcare-12-00534-t002:** Outcomes of selected studies.

No	Author	Primary Outcomes	Secondary Outcomes	Measurement Scales	Exp. Group M (SD) or *n* (%)	Con. Group M (SD) or *n* (%)	t or F or r	*p*
A1	Bibault [19]	① Quality of information	② Answer rate	① EORTCQLQ ② *n* (%)	① 2.89 ② 49 (69%)	① 2.82 ② 46 (64%)	① - ② -	① <0.001 ② -
A2	Greer [20]	① Anxiety ② Depression	③ Negative emotion ④ Positive emotion ⑤ Usage time	①–④ PROMIS ⑤ Minute	① 61.9 ± 7.7 ② 59.1 ± 9.2 ③ 1.5 ± 0.9 ④ 2.5 ± 1.0 ⑤ 12.1 ± 7.1	① 63.3 ± 5.5 ② 57.7 ± 6.1 ③ 1.6 ± 0.6 ④ 2.3 ± 0.8 ⑤ 18.1 ± 8.6	① 0.41 ② 0.09 ③ 0.01 ④ 0.07 ⑤ -	① 0.09 ② 0.77 ③ 0.97 ④ 0.82 ⑤ -
A3	Maeda [3]	① Anxiety	② Fertility knowledge ③ Intention of preconception	① STAI ② CFKS-J ③ Survey	① 43.2 ± 9.5 ② - ③ 68.7 ± 23.0	① 47.5 ± 9.5 ② - ③ 76.4 ± 18.4	① - ② - ③ -	① <0.001 ② 0.001~0.005 ③ <0.001
A4	Fitzsimmons-Craft [21]	① Eating disorder risk	② Internalization ③ Eating disorder ④ Depression ⑤ Anxiety	① WCS ② SATAQ-4R ③ EDE-Q ④ PHQ-8 ⑤ GAD-7	① 60.80 ± 20.55 ② 15.35 ± 3.94 ③ 2.77 ± 1.34 ④ 11.09 ± 6.42 ⑤ 10.40 ± 6.14	① 63.99 ± 17.30 ② 16.11 ± 3.84 ③ 3.03 ± 1.24 ④ 11.67 ± 5.55 ⑤ 10.96 ± 5.51	① −0.45 ② −0.21 ③ −0.38 ④ −0.26 ⑤ −0.11	① <0.001 ② 0.001 ③ <0.001 ④ <0.001 ⑤ 0.09
A5	Chung [4]	① Satisfaction	② Usability ③ Ese of use ④ Ease of learning	① SAT ② USE ③ EOU ④ EOL	① - ② - ③ - ④ -	① - ② - ③ - ④ -	① r = 0.97 ② r = 0.89 ③ r = 0.32 ④ r = 0.95	① <0.001 ② <0.001 ③ 0.24 ④ <0.001
A6	Yam [5]	① Acceptability	② Feasibility ③ Knowledge ④ Interaction	① - ② - ③ -	① 100% ② 97% ③ 83% ④ 96%	① - ② - ③ - ④ -	① - ② - ③ - ④ -	① - ② - ③ - ④ -
A7	Al-Hilli [22]	① Satisfaction	② Knowledge	① Median ② Median	① 30 (6–30) ② 11 (8–13)	① 30 (24–30) ② 12 (8–14)	① - ② -	① 0.19 ② 0.09
A8	De Filippo [23]	① Depression	② Gender attitudes ③ IPV exposure ④ Unhealthy relationships	① PHQ-2 ② GRS ③ WHO ④ VAS	① 17% ② 20.06 ③ 62% ④ 0.62	① 6.9% ② 19.56 ③ 55% ④ 0.55	① - ② - ③ - ④ -	① <0.01 ② <0.01 ③ >0.05 ④ <0.001
A9	Mane [24]	① Usability	② Use rate	① % ② %	① 61.76% ② 24.27%	① - ② -	① - ② -	① - ② -
A10	Tawfik [6]	① Physical effect ② Psychological effect ③ Distress	④ Effectiveness of self-care behavior ⑤ Usability	①–③ MSAS ④ SCBD ⑤ CUQ	① 1.37 ± 0.30 ② 1.42 ± 0.30 ③ 1.80 ± 0.93 ④ 2.42 ± 0.49 ⑤ 49.94 ± 5.64	① 2.77 ± 0.21 ② 2.79 ± 0.21 ③ 3.00 ± 0.30 ④ 1.81 ± 0.44 ⑤ -	① 97.0 ② 62.13 ③ 80.26 ④ 20.03 ⑤ -	① <0.001 ② <0.001 ③ <0.001 ④ <0.001 ⑤ -

CFKS-J = Japanese version of the Cardiff Fertility Knowledge Scale; Cont = control group; CUQ = Chatbot Usability Questionnaire; EDE-Q = Eating Disorder Examination Questionnaire; EOL = ease of learning; EORTCQLQ = questionnaire developed to assess the quality of life of cancer patients; EOU = ease of use; Exp = experimental group; GAD = generalized anxiety disorder; GRS = Gender Relation Scale; MSAS = Memorial Symptoms Assessment Scale; PHQ = Patient Health Questionnaire; PROMIS = Patient-Reported Outcome Measurement Information System; SATAQ-4R = Sociocultural Attitudes Toward Appearance Questionnaire—4; SAT = satisfaction; SCBD = self-care behavior diary; STAI = State–Trait Anxiety Inventory; USE = usability; VAS = Violence Assessment Scale; WCS = Weight and Shape Concern; WHO = World Health Organization multi-country study instrument.

## Data Availability

Not applicable.

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
