# Peer review of "The Effects of Artificial Intelligence Chatbots on Women’s Health: A Systematic Review and Meta-Analysis"

_healthcare, 2024, doi:10.3390/healthcare12050534_

Round 1
Reviewer 1 Report
Comments and Suggestions for Authors
This systematic review and meta-analysis examine the impact of artificial intelligence (AI) Chatbot interventions on women's health outcomes. The study includes a selection of ten relevant studies published between 2019 and 2023, focusing specifically on experimental studies in women's health. Adhering to the Preferred Reporting Items for Systematic Reviews and Meta-Analyses (PRISMA) guidelines, the review evaluates the quality of the literature using the ROB 2 checklist. The findings reveal that AI Chatbots demonstrate positive effects on a range of health outcomes, including anxiety, depression, distress, healthy relationships, cancer self-care behavior, preconception intentions, risk perception of eating disorders, and gender attitudes. A meta-analysis of three studies indicates a significant anxiety reduction. The conclusion suggests that AI Chatbot interventions have the potential to revolutionize nursing interventions for women, improving overall health status and offering digital therapy. Although the paper is interesting, the following comments should be addressed.
-- To enhance the introduction, consider incorporating a brief statement emphasizing the prevalence and significance of women's health issues.
-- Highlight how AI Chatbots can address specific challenges faced by women, thereby underscoring the importance of focusing on women's health in the context of AI Chatbot interventions.
-- The authors should point out the major contributions of this paper by using 3 to 5 brief bullet points at the end of the Introduction section, right before the last paragraph.
-- The structure of arguments needs to be improved. At the end of the introduction part, you should have a section plan (for example section 2 discusses... and section 3 gives...).
-- To strengthen the literature review, expand the discussion to include existing studies that have explored the effectiveness of Chatbots in healthcare interventions for specific populations.
-- Provide the discussion of more recent publications including ‘MIMONet: multi-input multi-output on-device deep learning’, ‘Association between type 2 diabetes mellitus and multiple myeloma: Fact or fiction’, ‘AI in education: cracking the code through challenges: a content analysis of one of the recent issues of educational technology and society (ET&S) journal’, ‘Pimbot: Policy and incentive manipulation for multi-robot reinforcement learning in social dilemmas’, and ‘Features requirement elicitation process for designing a Chatbot application’.
-- The paper provides a clear description of the study design and search strategy. However, consider including additional details regarding the inclusion and exclusion criteria employed for study selection.
-- To enhance clarity, consider organizing the results based on the specific health outcomes addressed by the Chatbot interventions.
-- Explore the enduring impact of AI Chatbot interventions on women's health outcomes over extended periods.
-- Examine the integration of AI Chatbots into existing healthcare systems to enhance the delivery of women's health services.
-- Conduct comparative studies to assess the effectiveness of AI Chatbots compared to other intervention modalities, such as face-to-face counseling, telehealth, or mobile applications.
-- Extend research on AI Chatbot interventions for women's health to diverse global populations.
-- To further enrich the discussion, address potential limitations of the included studies, such as sample size, study duration, and generalizability.
Comments on the Quality of English LanguageMinor editing of the English language is required.
Author Response
Your expertise and thorough analysis have not only helped me to improve the clarity and depth of my research but also guided me in addressing the critical points that I had initially overlooked. It is evident that your suggestions come from a place of extensive knowledge and a genuine desire to contribute to the advancement of our field. For this, I am truly thankful.
I have taken your feedback to heart and am in the process of revising my article accordingly. Your detailed comments have provided me with a clear direction on how to refine my arguments and strengthen my evidence, ensuring that my final manuscript will be of a much higher quality. It is reviewers like you who elevate the standards of scientific discourse and drive the collective progress of our research community.
Please accept my deepest appreciation for your contribution to my work. I am inspired by your dedication to fostering rigorous academic inquiry and am grateful for the opportunity to learn from your expertise. I hope that my revised article will reflect the high standards you uphold and make a meaningful contribution to our field.
Thank you once again for your invaluable support and guidance. I look forward to the possibility of engaging with you in future endeavors and continuing to benefit from your profound knowledge and insight.

Reviewer 2 Report
Comments and Suggestions for Authors
Authors need to mention which the research questions are addressed.
Which type of research study has been conducted? Which sampling technique is used?
Authors has taken existing chatbots for analysis. It can also be mentioned that how basic AI chatbot works?
Comments on the Quality of English LanguageLittle modification can be done.
Author Response

(The authors gave the same response as above.)

Reviewer 3 Report
Comments and Suggestions for Authors
The systematic review article is well-designed and attractively written.
The study aims to investigate the effects of artificial intelligence chatbot interventions on women's health outcomes.
The study has focused extensively on ten significant scientific research papers published between 2019 and 2023, delving into experimental studies that center around artificial intelligence chatbot interventions in women's health. I have not encountered such a compilation specifically focused on chatbots before.
The study has demonstrated that artificial intelligence chatbot interventions have positive effects on women's physical, physiological, and cognitive health outcomes. This could potentially alter nursing interventions as a form of digital therapy to improve the health status of the female population and provide social support.
The authors have implemented a solid methodology. However, the review extensively focuses on the advantages of chatbots. It would be beneficial to also mention their disadvantages.
The results of the study indicate positive effects of artificial intelligence chatbot interventions on anxiety.
Figure 3 should be presented in a more explanatory manner; it is unclear what the shapes presented in Figure 3 mean. It should be specified what the shapes represent.
Author Response

(The authors gave the same response as above.)

Reviewer 4 Report
Comments and Suggestions for Authors
The Effects of Artificial Intelligence Chatbots on Women’s Health: A Systematic Review and Meta-Analysis
The document presents a systematic review and meta-analysis of the impact of artificial intelligence (AI) chatbot interventions on women's health outcomes.
The limitations of the review paper in the study are as follows:
1) Could you provide evidence on how various outcome measures, ROB 2 and robvis quality scores, and meta-analysis results with an anxiety effect size of -0.30 influence the perceived effectiveness and reliability?
2) Please explain how the variations in intervention duration, settings, study designs, and participant demographics affect the outcomes and generalizability.
3) How does limiting the search to English and Korean publications affect the comprehensiveness of the review?
4) I recommend that you add the most recent publications to your article, which will further enrich your article:
a) Enhancing Medical Image Denoising with Innovative Teacher–Student Model-Based Approaches for Precision Diagnostics. Sensors 2023, 23, 9502. https://doi.org/10.3390/s23239502.
b) Deep learning-driven diagnosis: A multi-task approach for segmenting stroke and Bell's palsy, Pattern Recognition. https://doi.org/10.1016/j.patcog.2023.109866.

Comments on the Quality of English LanguageNone
Author Response

(The authors gave the same response as above.)

Reviewer 5 Report
Comments and Suggestions for Authors
Although stated in the conclusions that research included articles until October 2023, the time span should however be limited to 2019-2022.
No mention of possible pandemic effects on adoption and use of chatbot is present, although Gabrielli and al.'s paper (citation n.24) consider it.
Comments on time span inclusion criteria for findings since 2011 (and the introduction of SIRI i presume) would be recommended. Chatbot performance from 2011 and 2023 can't be compared, unless further criteria be introduced.
Queries based on "instant messaging" and "text messages" should have been avoided as can't be directly related to the premises of the research.
Further considerations on such different cultures and countries included in the results should be mention.
Author Response

(The authors gave the same response as above.)

Round 2
Reviewer 1 Report
Comments and Suggestions for Authors
The authors have carefully addressed all the reviewing comments raised in the first round of review. It's my pleasure to recommend acceptance of the current version.
Author Response
I am grateful for your insightful feedback and the opportunity to improve my manuscript. Thank you very much for your time and consideration.
Reviewer 4 Report
Comments and Suggestions for Authors
I agree
Comments on the Quality of English LanguageI agree
Author Response

(The authors gave the same response as above.)

Reviewer 5 Report
Comments and Suggestions for Authors
Search period could have been extended to include all 2023 in order to harvest 5 complete years.
A "conclusions" paragraph is missing.
The study about cognitive load (citation n.21 cfr. https://arxiv.org/abs/2111.01400) doesn't tackle womens' mental health, but enterprise productivity in B2B settings; however conclusions are not definitive and mention also "frustrating experiences" with chatbots.
Author Response

(The authors gave the same response as above.)
